# Practitioners' perspectives on implementation of acute virtual wards: A scoping review

**Theresa Sunny[1], Nandakumar Ravichandran[ID][1]\*, John Broughan[2], Geoff McCombe[1], Sheila Loughman[1], Kenneth McDonald[1,3], Neasa Starr[4], Walter Cullen[1]**

1 School of Medicine, University College Dublin, Dublin, Ireland, 2 Clinical Research Centre, School of Medicine, University College Dublin, Dublin, Ireland, 3 Department of Cardiovascular Medicine, St Vincents University Hospital, Dublin, Ireland, 4 Department of Cardiovascular Medicine, University Hospital Limerick, Limerick, Ireland

\* nandakumar.ravichandran@ucd.ie

## Abstract

Virtual wards provide a promising alternative to traditional 'bedded care' by facilitating early discharges and delivering acute care at home. They focus specifically on patients needing acute care, which would traditionally necessitate an in-hospital stay. Understanding practitioners' beliefs and attitudes is crucial for successful implementation and operation of Virtual wards. This scoping review explores practitioners' perspectives on the implementation of virtual wards. A total of 18 studies were included in the final analysis from the 201 studies identified initially through searches in PubMed, Cochrane, CINAHL, and Embase databases (2015–2024) following PRISMA Extension for Scoping Reviews (PRISMA-ScR) guidelines. Thematic analysis was conducted using Braun and Clarke's framework to identify key insights. Thematic analysis revealed key themes related to implementation, quality of care, technology, training, and awareness. These themes highlight the challenges influencing the adoption and considerations for the operational success of virtual wards. Virtual wards demonstrate significant potential for delivering acute care efficiently and sustainably. However, challenges related to service design, patient safety, technology integration, and workforce training must be addressed to ensure their successful implementation and long-term efficacy.

## Author summary

Virtual wards offer a transformative approach to delivering acute care by enabling early hospital discharges and providing treatment at home, reducing reliance on traditional hospital beds. This scoping review examines practitioners' perspectives on the implementation of virtual wards, as their attitudes and beliefs are pivotal to successful adoption. The review analyzed 18 studies from

**Data availability statement:** All data are in the manuscript.

**Funding:** The first author, TS, a medical student, received funding support through the Dr. Mary J. Farrell Scholarship for this work. No other authors (NR, JB, GM, SL, KM, NS, and WC) received any funding support for this study. The funders had no role in study design, data collection and analysis, decision to publish, or preparation of the manuscript.

**Competing interests:** The authors have declared that no competing interests exist.

201 studies identified in PubMed, Cochrane, CINAHL, and Embase databases (2015–2024), following PRISMA-ScR guidelines. Thematic analysis using Braun and Clarke's framework revealed five key themes: implementation challenges, quality of care, technology, training, and awareness. While virtual wards hold immense potential for efficient and sustainable healthcare delivery, significant barriers must be addressed, including service design, patient safety, digital integration, and workforce training. Understanding these practitioner-identified challenges is critical to optimizing the adoption, functionality, and long-term success of virtual wards, ensuring they meet patient needs and maintain high standards of care.

## Introduction

Healthcare systems worldwide are under pressure, largely as a result of aging populations and the resulting demand for healthcare services [1]. In Ireland, there is an increased risk of hospital admission largely due to demographic growth, population ageing and chronic disease, as well as other factors including seasonal surges in respiratory illnesses and health service capacity limitations [2]. These factors combined lead to more emergency department (ED) presentations and hospital overcrowding, with hospitals working at full capacity. This is culminating in sustained and recorded pressure on ED services. The associated ED 'overcrowding' is linked to the inefficient use of resources, increased patient risk and a challenging work environment for staff [3]. The current capacity of hospital inpatient systems cannot keep up with these demands and shifting demographics [1]. Acute hospital admission avoidance interventions such as Virtual wards (VWs) or Hospital at Home (HaH) offer alternatives to traditional 'bedded care' to prevent avoidable admissions (step-up) or support early discharges (step-down) [1,4].

In current NHS documentation, a VW is defined as: 'a safe and efficient alternative to NHS bedded care that is enabled by technology…virtual wards support patients who would otherwise be in hospital to receive the acute care, monitoring and treatment they need in their own home' [4,5]. VWs allow key elements of acute care traditionally delivered in hospitals to be provided at home including intravenous treatments, MDT assessment and rehabilitation, physiological monitoring using digital technologies, and senior clinician decision making [6]. A range of common clinical presentations, including frailty and heart failure exacerbations, have also been shown to be amenable to virtual care approaches [6,7]

There remains a persistent shortage of acute hospital beds to meet the current demand [8]. Given this context and the development of VWs, there is a need to review the current literature exploring the acceptability and likely effectiveness of VWs.

Understanding practitioners' beliefs and attitudes is essential to developing effective strategies to enhance healthcare practices. Research has emphasized that studying the attitudes of those directly involved in implementation is a crucial

component of the process [9,10]. Additionally, successful implementation projects recognize the importance of an interdisciplinary team, with clinicians being key stakeholders as end users. Hence, our scoping review focuses on practitioners' perspectives in order to include diverse stakeholder insights, which are crucial for collaborative implementation and improved service delivery [11]. This scoping review aims to explore and report on the challenges, considerations and likely effectiveness as identified by practitioners in implementing VWs.

## Methods

In order to help guide practice, policymaking, and research, scoping studies map the literature on a certain issue or study area and offer a chance to identify significant ideas, research gaps, and forms and sources of evidence [12]. In this context, a scoping review methodology was adopted, as this approach was considered optimal to provide an overview of a rapidly evolving topic, such as acute virtual wards. This scoping review was conducted between May and July 2024. The review framework followed Arksey and O'Malley's [13] six-step methodological framework with additional recommendations provided by Levac et al (2010) [14]. This framework involves identifying the research question, conducting comprehensive search, selecting relevant studies, charting the data, and summarising the findings. The six stages of the scoping process are described below.

### Stage 1: Identifying the research question

This scoping review aimed to determine the challenges, considerations and likely effectiveness of implementing acute virtual wards from the perspectives of practitioners involved in VWs. The following research question was formulated:

"What are practitioners' perspectives on the implementation of acute virtual wards?"

### Stage 2: Identifying relevant studies

A preliminary search of key databases and grey literature was performed, using multiple search terms to create a reading list. From this list, key words were identified.

The online databases searched during the literature review were PubMed, Embase, CINAHL and Cochrane Library. Several sources that met the predefined inclusion criteria were incorporated from other sources to ensure a comprehensive review.

Searches covered publications from 2015 to 2024 and were limited to English. A sensitive search strategy was implemented, and the search syntax is detailed below.

(("acute care facility*" OR "emergency medicine" OR acute) AND ("virtual ward*" OR "hospital at home" OR "virtual care") AND (physician* OR doctor* OR clinician* OR "healthcare provider*") AND (perspective* OR viewpoint* OR attitude* OR experience* OR opinion* OR efficacy OR implementation OR satisfaction))

### Stage 3: Study selection

The initial search generated 201 results and were compiled into a preliminary reading list. The PRISMA Extension for Scoping Reviews (PRISMA-ScR) flow diagram outlines the study selection process (Fig 1).

Endnote 20 software was used to track and group studies, manage citations and remove duplications. The inclusion criteria (see Table 1) were broad to include a range of articles, consistent with scoping reviews methodology. Literature was included irrespective of study design. Covidence, a web-based collaboration software platform, was used for screening the studies. Two reviewers independently screened each article. After an initial screening of titles and abstracts by the first reviewer, the second reviewer conducted a secondary screening. Full texts were retrieved for studies meeting the inclusion criteria or in cases where suitability was uncertain. If a study seemed to fulfil the inclusion criteria but data was insufficient or involved the wrong comparator, it was excluded. This scoping review was completed with a final selection of 18 papers, upon which both reviewers agreed.

PLOS Digital Health

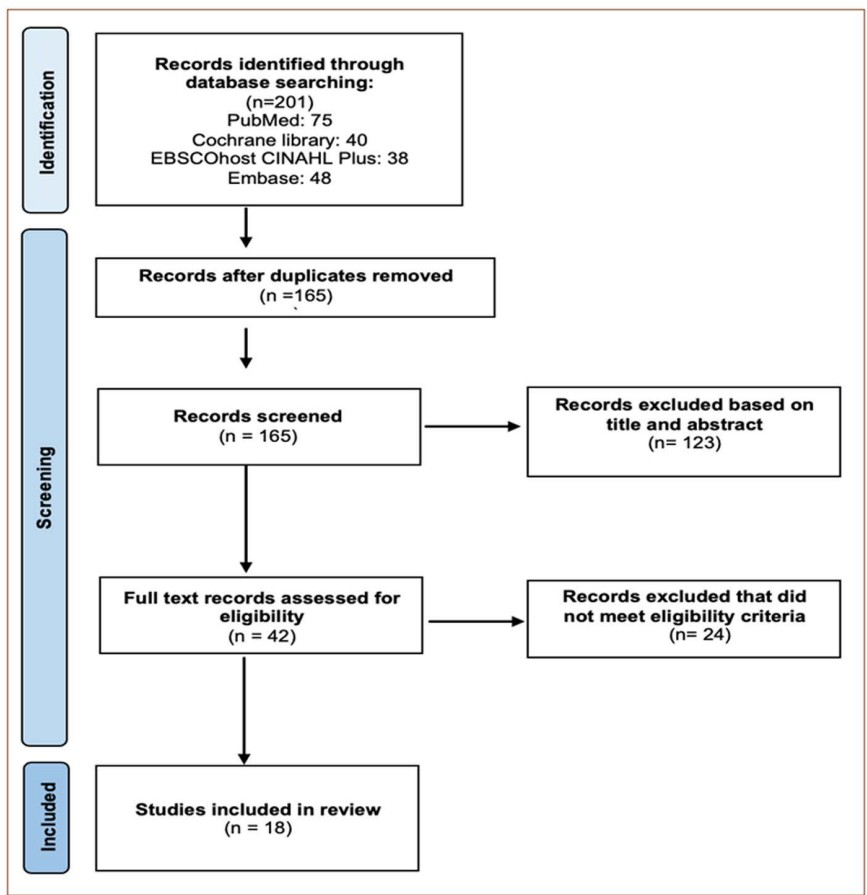

**Fig 1. PRISMA-ScR flow diagram.**

**Table 1. Inclusion and exclusion criteria.**

|  | Inclusion criteria | Exclusion criteria |
|---|---|---|
| **Population** | Practitioners, Staff, MDT, Healthcare providers, Key stakeholders | Caregivers, Patients |
| **Context** | Focus acute virtual wards, acute HaH, urgent virtual care | Focus on telemedicine, non-acute virtual care, in-hospital care |
| **Concept** | Perspective, experiences, recommendations of practitioners, MDT, healthcare providers | No mention of practitioner, MDT, healthcare provider perspectives, experiences or recommendations |
| **Type of evidence** | Articles written in English, published between 2015–2024, reviews | Protocols, conference abstracts |

## Stage 4: Charting the data

Once all exclusion criteria were applied, the data from the remaining studies were charted. We summarized them in a table format to allow for comparison and thematic analysis under the following headings of:

- Author(s), year of publication

- Study design

- Study population

- Location

- Intervention

- Aim of study

- Outcome/major finding

**Stage 5: Collecting summarizing and reporting results**

Extracted data were collected, presented and reported in the results section (see Table 2). The data were coded using NVivo by two researchers (TS and NR) under the supervision of WC, with input from JB, GM, and SL, who collaborated in refining themes. The study team (KM and NS) also included physicians involved in the implementation of AVWs in Ireland. To ensure consistency, TS and NR independently coded the data before discussing and refining themes with the broader team. Discrepancies were resolved through group discussion and consensus. The analysis followed an inductive approach. Key themes were identified using thematic analysis as informed by Braun and Clarke [33].

## Results

Initial searches of PubMed, Embase, Cochrane Library and CINAHL yielded 201 results. Following duplicate removal and reviewing of titles and abstracts, 42 studies were then followed with a full-text review. The search, identification and selection processes are summarized in the PRISMA ScR flow diagram (Fig 1). The following 18 studies were selected for the final inclusion.

**Study characteristics**

The 18 studies included were from six countries including Australia (n = 1), Canada (n = 5), Italy (n = 1), Spain (n = 1), the United States (n = 4) and United Kingdom (n = 5) and one study comprising Australia and United Kingdom and adopted a range of methodologies. The study designs included qualitative studies (n = 4), systematic reviews (n = 2), organizational studies (n = 1), retrospective studies (n = 3), descriptive studies (n = 3), scoping reviews (n = 1), rapid reviews (n = 1), multi-methods studies (n = 1), article reviews (n = 1), and study reports (n = 1). The studies used a variety of terminologies to describe acute VWs. Among the most used were virtual ward, virtual urgent care and hospital-at-home.

The thematic analysis from the scoping review identified key themes: **implementation, quality of care, technology, and training/awareness.**

Research showed high job satisfaction and engagement among multidisciplinary staff (e.g. GPs, nurses, admin staff, etc.) involved in VW compared to traditional hospital settings [15]. Healthcare staff involved in VWs expressed enthusiasm for the continued development of VWs, highlighting the meaningful impact they felt in delivering care [21]. Practitioners and physicians also found value in virtual care models, noting that they were able to effectively support patients, which led to increased motivation and satisfaction with their work [28]. Furthermore, patients, caregivers, and healthcare professionals consistently reported positive health outcomes and satisfaction with VW services, reinforcing the perceived benefits and effectiveness of these models in healthcare delivery [20,31]. However, several challenges were identified that may influence their effectiveness.

**1. Implementation. Service design:** AVWs provide an efficient alternative to in-person admission but face implementation challenges [15] in staffing models and patient inclusivity—addressing socioeconomic and language barriers [20,22]. Staffing models varied, with some using disease-specific virtual units led by specialty-trained practitioners [34], while others relied on on-call emergency physicians or integrated intermediate triage steps involving nurses, physician assistants, or nurse practitioners [20] (Fig 2). Non-clinical staff, such as hospital administrators, reduced ED workload by managing registration, patient transfers, and technological support [24,25,29].

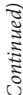

**Table 2. Description of studies included in scoping review.**

| Author(s), Year | Journal | Location | Population | Intervention | Aim/Topic | Study design | Major Findings |
|---|---|---|---|---|---|---|---|
| Anwar et al. (2024) [15] | European Geriatric Medicine | Scotland | 23 staff | Virtual ward (HaH) | Understand challenges and opportunities of HaH | Qualitative interview study | High job satisfaction with concerns about staff safety, service awareness, and sustainability |
| Cabrera López et al. (2022) [16] | Journal of Paediatric and Child Health | Spain | 833 paediatric patients | Paediatric Virtual ward (HaH) | Describe HaH experience | Retrospective observational study | Majority referred from inpatient wards, minimal direct ED admissions. |
| Fawcett et al. (2022) [17] | Cureus | United States | 65 paediatric emergency physicians | Telemedicine – virtual video calls | Assess comfort in providing urgent adult care via telemedicine during COVID-19 | Retrospective pre-post cross sectional survey | Increased comfort with higher patient volumes, treatment algorithms, group support, and real-time backup. |
| Gonçalves-Bradley et al. (2017) [18] | Cochrane Database of Systematic Reviews | UK | 32 trials | Early discharge Virtual ward (HaH) | Assess effectiveness and cost of early discharge HaH vs. inpatient care | Systematic review | Reduces hospital stay and institutionalization, no impact on health outcomes/readmissions |
| Grant et al. (2024) [19] | Canadian Journal of Emergency Medicine | Canada | Members of Canadian association of emergency physicians | Virtual care in ED | Highlight opportunities in emergency medicine using virtual care | Descriptive study | Mixed perceptions of digital Emergency medicine |
| Hall et al. (2022) [20] | Frontiers in Digital Health | Canada | 13 emergency physicians and researchers | Virtual urgent care | Describe design, facilitators, barriers of VUC pilot programs | Descriptive study | Facilitators: local champions to guide program delivery, provincial funding support, patient involvement, multipronged marketing. Barriers: behaviour change strategies, IT quality, data collection. |
| Jung et al. (2023) [21] | Medical Care Research and Review | United States | 17 practitioners and staff | Virtual observation unit in ED | Describe experiences of implementing COVID-19 Virtual Observation Unit (CVOU) | Qualitative study | Increased practitioner-staff interaction, patient/family participation, attention to nonmedical factors, expanded roles |
| Lawrence et al. (2023) [22] | Frontiers in Digital Health | Australia | 9 practitioners | Virtual ward (HaH) | Measure feasibility, acceptability, safety, impact of virtual care for children with COVID-19 infection requiring hospital-level care | Retrospective study | Video consultations most useful, followed by remote monitoring |
| Liu et al. (2019) [23] | AMIA Annual Symp Proc | United States | 25 providers | virtual urgent clinic | Understand VUC physician characteristics and prescribing patterns | Descriptive and multivariate analysis study | Variations in prescribing patterns could compromise virtual care quality |
| Mehta et al. (2024) [24] | Canadian Journal of Emergency Medicine | Canada | 1 EM resident and 7 emergency physicians | Virtual urgent care | Characterise current state, key infrastructure, and recommend improvements for VUC | Scoping review | VUC should have certified providers, accessible video platforms, ensure equity, timely in-person care, integrated networks, and public funding |

*(Continued)*

**Table 2.** (Continued)

| Author(s), Year | Journal | Location | Population | Intervention | Aim/Topic | Study design | Major Findings |
|---|---|---|---|---|---|---|---|
| Parks et al. (2021) [25] | Australia Journal of Rural Health | Wales | Practitioners and emergency transport providers | Virtual, Coordination, Access, Referral and Escalation (vCare) | Outline challenges, initiatives, outcomes of vCare | Report | Delegated authority improved communication and referral patterns, eased ED burdens |
| Patel et al. (2021) [26] | Global Journal of Qual Saf Healthcare | United States | Patients, physicians, healthcare management | Virtual ward (HaH) | Provide Perspectives on HaH | Review | Concerns about care efficiency, time constraints, financial aspects |
| Sheppard et al. (2021) [27] | Cochrane Database of Systematic Reviews | UK | 4 RCT | Virtual ward (HaH) | Assess home-based end-of-life care impact vs. inpatient/hospice care | Systematic review | Need for additional caregiver support |
| Shuldiner et al. (2023) [28] | JMIR Formative Research | Canada | 14 healthcare providers | Virtual emergency department | Assess model acceptability, quality, access and continuity of care | Multimethod study | Well-received by physicians, some concerns about underuse of expert skills |
| Shuldiner et al. (2022) [29] | JMIR Human Factors | Canada | 14 physicians | Virtual emergency department | Understand integration of virtual ED with in-person operations | Qualitative study | Mixed physician experiences |
| Tibaldi et al. (2019) [30] | International Journal of Clinical Practice | Italy | 13 senior nurses and one nursing manager | Virtual ward (HaH) | Provide insights into the organisational structure, protocols, and multidisciplinary approach | Organisational study | Close collaboration among stakeholders is key to success |
| Wallis et al. (2024) [31] | Cochrane Database of Systematic Reviews | UK and Australia | 52 studies | Virtual ward (HaH) | Identify and synthesise factors influencing HaH implementation | Qualitative evidence synthesis (review) | Requires early policy development, stakeholder engagement, efficient processes, effective communication, skilled workforce |
| Westby et al. (2024) [32] | Age and Ageing | UK | 18 documents and experience of practitioners | Virtual ward | Identify VW models, explore effectiveness for people with frailty | Rapid review | Guarantee of patient safety and benefit needed |

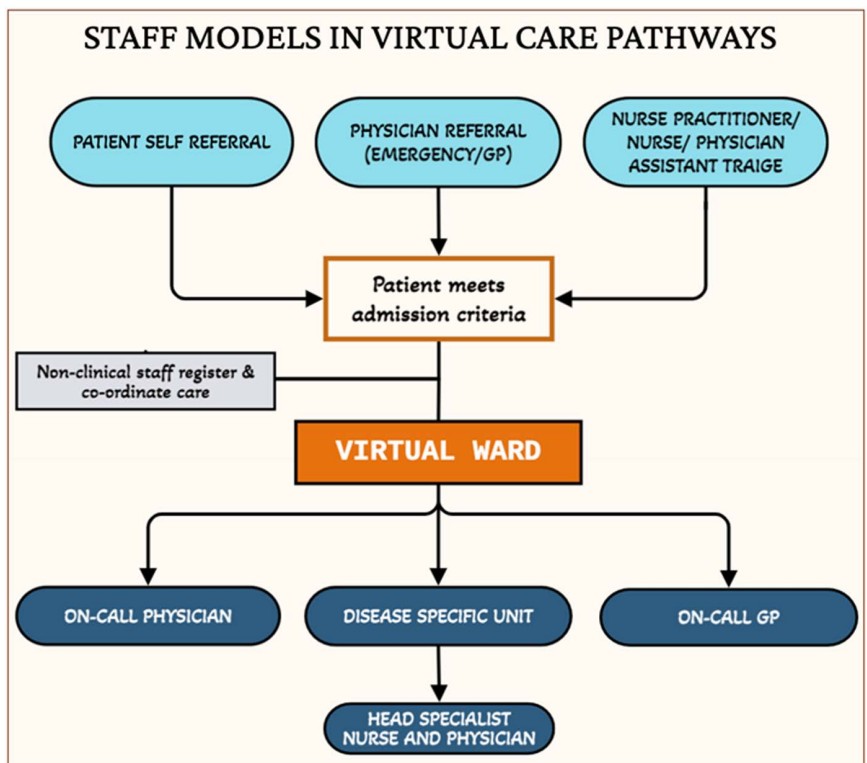

**Fig 2. Staff models in virtual wards.**

Resource allocation (human resources and funding for staffing and equipment) was a common frustration [31], especially during the pandemic, when additional in-person hours already strained staff [20]. Recruitment challenges were particularly notable in rural settings, though voluntary staff recruitment was more effective, fostering innovation and engagement [31]. Depending on the local context, this model may be more beneficial for out of hours or overnight shifts [24,27]. A 24/7 on-call VW physician can ensure timely response to acute medical crises [26]. Logistical inefficiencies, like transporting blood samples to hospital labs, reduced patient contact time; incorporating point-of-care testing offered a solution, expediting decisions within single visits [15]. Higher patient volumes reported in models where hospital administrators actively supported implementation, as evidenced by resource allocation, policy facilitation, and engagement in service development. Additionally, integrating patient perspectives and fostering broad patient awareness contributed to increased patient uptake [20].

**Prioritization of acute patients:** AVWs offer flexibility in prioritizing patients based on acuity [15]. Self-screening models reduced redundancy in history-taking and interaction time [20], but prioritizing more unwell patients limited the feasibility of combined visits, while split visits involving nurse practitioners and physicians, though effective for acuity-based triage, increased consultation times and resource costs [15,20]. Physicians expressed concerns about virtual EDs being misused for faster specialist access [29], highlighting the need for appropriate referrals [24,28]. Defining appropriate presentations for virtual urgent care remains challenging, as low-acuity patients may still require in-person care [24]. Prioritization depends on clinical judgment and tools like frailty and risk prediction scores, which help optimize care and reduce hospitalizations [32].

**Sustainability:** AVWs offer system-wide benefits, such as reduced emissions, cost savings, and shorter hospital stays [15,24,31]. VWs has the potential to reduce travel-related emissions of patients and staff by promoting greener models of healthcare with reduced hospital stays and alternative care models, allowing to deliver healthcare in the future of challenging weather events that might have impact on patient and staff mobility. However, concerns about climate impact from vehicle use persist, with recommendations for electric vehicles [15]. Financial sustainability remains a challenge due to inadequate funding for staff, equipment and home care services requiring public funding and regional support. Standardized outcome measures and patient-reported data are crucial for evaluating the impact of VWs and guiding funding decisions [20,24] (Table 3).

 **2. Quality of care. Staff and patient safety:** Patient safety issues, including pain management and mobility at home, can be alleviated with appropriate equipment delivery [31]. In a virtual care model, home visits are sometimes necessary for tasks such as physical assessments, equipment setup, blood work or urgent interventions that cannot be managed remotely. Staff safety concerns primarily revolved around home hazards and lone worker visits, where a single healthcare professional conducts these visits without immediate on-site support. Although lone worker alarm devices increased confidence among both providers and healthcare organizations, their limitations (e.g., not directly escalating to police) remain a concern regarding overall safety. Technology such as GPS tracking and recording devices may enhance safety [15]. However, effective implementation of a virtual care program should balance these benefits with concerns about privacy and the heightened sense of surveillance among service providers, ensuring that both physical safety and digital security of patients and staff are upheld.

**Consultation:** While many practitioners supported the VW model, concerns about care quality and under utilization of ED skills initially deterred referrals [28,29]. Physicians expertise in managing undifferentiated patients can be impactful in the continuation of quality assessments in this model [19]. Physicians who found the VW valuable reported benefits to patients and satisfaction from using their knowledge and skills to determine the need for an in-person visit, diagnostic tests or specialist referrals [20,28]. The virtual model sometimes limited physicians' ability to use key ED resources, leading to feelings of unfulfillment [28]. Additionally, some physicians felt patient issues were not fully resolved by the end of consultation, a challenge that can arise in any care model but was exacerbated by the virtual setting, causing frustration [29]. Video-based telehealth, supported by paramedics providing in-home care (e.g., vital signs, medications) [21], visual assessment supported by real-time oximetry data [22], enhanced capabilities and practitioner confidence [28]. Virtual care

**Table 3. Challenges, considerations and effectiveness of service design, prioritization of acute patients and sustainability in VWs.**

| | Challenges | Considerations | Effectiveness |
|---|---|---|---|
| **Service Design** | • Difficulty recruiting staff<br>• Addressing diverse patient cohort needs (socioeconomic factors, language barriers)<br>• Lack of resource allocation<br>• Potential inefficiency- transporting blood samples | • Incorporate non-clinical staff<br>• Utilizing 24/7 on-call physician<br>• Point of care blood testing<br>• Disease-specific virtual care units | • Providers found service better staffed than usual discharge services<br>• Similar workloads reported by GPs compared to in-person services<br>• Enthusiasm and openness to innovation seen in voluntary staff<br>• Higher patient volumes seen in models with supportive hospital administrators and integrated patient perspectives |
| **Prioritization of acute patients** | • Repetition of effort in triage models<br>• Incorrect use of services | • Early triage of illness acuity<br>• Use of frailty/hospital risk prediction tools | • Patient self-screening models reduced redundancy<br>• Combined visits may not always be possible when prioritizing more unwell patients<br>• Triage models effectively managed patient acuity |
| **Sustainability** | • Possible concern- climate impact of vehicles<br>• Funding<br>• Lack of standardized outcome measures and impact assessments | • Use of electric vehicles<br>• Frequent reporting and data collection<br>• Patient reported outcome and experience measures | • Virtual urgent care models were less expensive and shorter in visit duration than in-person care [22] |

required practitioners to learn from patient-generated data without physical exams, increasing the need for patients to play an active role in their care rather than just lying in a hospital bed in the case of ED [21]. Staff appreciated the benefit of seeing patients in a home environment, helping them to understand the interplay between home environment and symptoms better, providing additional context during care interactions [20,21,28]. Virtual consultations offered longer, more personalized interactions (30–45 minutes) compared to traditional ED settings of "10 minutes interactions" but often felt time-consuming, especially for introducing the model and screening patients. These consultations extended care beyond ED confines and allowed providers to address chronic and non-medical issues, offering additional value despite time constraints [21].

**Medication management:** Virtual care settings revealed differences in prescription practices, with emergency physicians being more cautious than primary care providers, potentially impacting care quality [23]. Polypharmacy and inappropriate medication use among home-care patients remain challenges [26,32]. The lack of available testing may also lead to over-prescription of antibiotics (27). The AVW model provides opportunities for personalized medication reviews at home, allowing for discussions on management, side effects, and adherence. These reviews can reduce adverse effects and improve patient outcomes [32] (Table 4).

  **3. Technology.  Methods of virtual monitoring:** A wide range of technology has been reported [20]. Virtual urgent care visits should use video platforms and web portals that are easy to access for patients and providers [24]. Video consultations, though effective for observing physical symptoms (e.g., rashes or injuries), were resource-intensive [22,24]. Lack of an integrated electronic health record (EHR) disrupted continuity between virtual and in-person care, requiring manual processes like faxing prescriptions and assessment notes [20,31]. An integrated EHR would allow for smooth transitions of care, referrals and prescriptions [24]. Digital inequality and access to technology as a barrier was identified in many studies, with some programs donating IT infrastructure to local shelters [20]. Privacy concerns require secure platforms conforming to national standards, with patient consent [24].

**Improved communication:** Effective communication between virtual and ED teams is essential to address staff isolation, fragmented decision-making, and redundancies [21,25,31]. Coordination through practices like "warm hand-offs" (detailed assessment notes) and structured discussions of risks and benefits of virtual versus in-person care were particularly helpful [16,20,31,32]. Overall, daily communication via emails among providers discussing cases, management and the availability of a physician to consult during a shift were found to be beneficial [17,32]. Staff reported that virtual care models provided additional time to share information with patients and colleagues, and highlighted the convenience and timeliness of virtual information exchange [21]. Reinforced team interactions in the virtual workspace encouraged co-learning and trust-building among practitioners, contributing to lower burnout and increased job satisfaction. Workflow designs that prioritize team interactions can further enhance these benefits [21,25,31,34] (Table 5).

**Table 4.  Challenges, considerations and effectiveness of staff and patient safety, consultation, medication management in VWs.**

| | Challenges | Considerations | Effectiveness |
|---|---|---|---|
| **Staff and Patient Safety** | • Pain management and mobility of patients assigned to home care<br>• Safety concern - Home hazards<br>• Potential concern – lone worker visits | • Appropriate home equipment delivery<br>• Staff use of safety alarm, recording or GPS devices | • Lone worker alarms increased staff confidence<br>• Home visits conducted by a single healthcare professional raised safety concerns |
| **Consultation** | • Provider concern - Under utilization of skills<br>• Time constraints<br>• Virtual examinations | • Incorporate paramedics to obtain vital signs and administer medication | • Paramedics' in-home care improved capabilities like obtaining vital signs and administering medication<br>• Expanded capabilities and visual assessments and real-time data (e.g., oximetry) improved practitioner satisfaction |
| **Medication Management** | • Different prescription patterns<br>• Polypharmacy | • Extra time for medication management discussion | • Personalized medication reviews increased adherence and reduced the risk of adverse effects |

**Table 5. Challenges, considerations and effectiveness on methods of virtual monitoring and improved communication in VWs.**

|  | Challenges | Considerations | Effectiveness |
|---|---|---|---|
| Methods of virtual monitoring | • Lack of united electronic health record between in-person and virtual care<br>• Inability to eFax prescriptions and notes<br>• Digital inequality - Access to cameras, high speed internet<br>• Privacy concerns | • Use of easy access video platforms and web portals<br>• Secure platforms – national standards | • Video consultations were resource intensive, but allowed the provider to conduct a facilitated physical exam<br>• Administrator assistance for technology issues was needed for smooth implementation<br>• Donations of IT infrastructure to local shelters helped mitigate digital inequality<br>• Practitioner satisfaction increased with a comprehensive and accurate system view of patient records |
| Improved communication | • Physical isolation, poor communication and isolated decision making for staff | • Use of a warm hand off<br>• Discussion with physician regarding benefits/risks of service<br>• Daily communication among providers | • Daily emails and the availability of a physician for consultations were beneficial<br>• Virtual urgent care model provided a safe setting for staff to ask questions and document collaboratively<br>• Increased time for providing patient information and communication with other providers<br>• Re-enforced interactions led to trust-building and co-learning among practitioners and staff |

**4. Training and awareness. Referral guidelines:** Clear eligibility and suitability criteria are crucial to ensure appropriate referrals [15]. In particular, identifying patients using a well-defined eligibility framework and clinical judgement was challenging for referrers in the early stages of implementation [31]. This led to a high number of misaligned referrals that did not meet service requirements. Regular updates on service capacity, shared daily with potential referrers, significantly increased referral volumes [15]. Additionally, direct collaboration with hospital specialists, particularly cardiologists, played a critical role in promoting referrals to VW [31]. This engagement included cardiologists providing clear referral criteria during multidisciplinary team meetings, conducting informational workshops for referrers, and offering ongoing case-based guidance to ensure appropriate patient selection. Services should develop clear inclusion and exclusion criteria to manage capacity, maintain activity and offer training sessions for appropriate referring.

**Education programs:** Virtual urgent care requires specific training for providers, including troubleshooting technology and conducting virtual exams [24]. The need for in-person care can also be unpredictable, requiring providers to understand local context and available pathways [24]. Implementing virtual urgent care requires staff training and role expansion beyond the normal scope of practice. Wallis et al. (2024) explored expanding nursing roles to increase home care capacity, requiring appropriate structures and policy changes [31]. To build capacity and comfort for this model in healthcare, a competency-based digital health curriculum or virtual training sessions supporting high-quality care, should be established for trainees, in-practice physicians and other healthcare providers [17,24]. Staff were open to the use of "step-by-step" training documents [21]. Patients and caregivers also expressed concerns about understanding the model, with written information alone being insufficient [31]. Therefore, communicating with patients to raise awareness for the new service is essential [20]. Cabrera et al. (2022) found caregiver training in therapeutic techniques key to home admission success [16]. This highlights the need for education programs targeted towards caregivers and patients [26]. These challenges highlight the need for additional training on home based care targeted towards professionals, patients and caregivers in order to improve care efficacy and enhance communication [26] (Table 6).

## Discussion

### Key findings

Virtual Wards may offer a promising alternative to in-patient care by providing acute care in patients' homes. Studies included in this review indicate high levels of satisfaction among healthcare staff and patients, highlighting the positive impact of VWs in delivering effective care. However, our scoping review also identifies several challenges and

**Table 6. Challenges, considerations and effectiveness on referral guidelines and education programmes in VWs.**

| | Challenges | Considerations | Effectiveness |
|---|---|---|---|
| **Referral guidelines** | • Misunderstanding suitability criteria – inappropriate referrals<br>• Low-capacity operation | • Sharing capacity updates<br>• Communication between specialties<br>• Develop suitability criteria to manage capacity<br>• Offer staff training for appropriate referring | • Sharing capacity updates to referrers daily by email increased referrals<br>• Engagement with hospital specialists promoted referrals to VWs |
| **Education programs** | • Challenges associated with working in a new environment<br>• Written information insufficient for patients | • Staff role expansion<br>• Use of a competency-based digital health curriculum and virtual training sessions<br>• Use of step-by-step training documents<br>• Caregiver training in therapeutic techniques<br>• Education program for patients and caregivers | • "Step-by-step" training documents were well-received by staff<br>• Education programs targeted towards caregivers and patients improved care and communication |

considerations associated with the implementation of acute virtual wards. These are organized around four key themes: implementation, quality of care, technology and communication, and training and awareness.

## Comparing with existing literature

**Implementation.** A central theme in our review was the need for robust service design that prioritizes acute patients and ensures sustainability. Our findings align with Norman et al. (2023), who identified a lack of guidance on key aspects of VW design, such as team roles and data protection [4]. Flexible staffing models were crucial, especially in remote settings, which supports recommendations from Gilhooly et al. (2019) to co-design adaptable services [35]. The involvement of patients and caregivers in service design was also emphasized as crucial for service acceptance, consistent with recent studies [36,37].

Effective prioritization and clear escalation pathways were key to VW success. However, Greene et al. (2024) noted challenges in transferring patients to in-patient care during ED overcrowding [38]. These findings align with a report from the Health Foundation, England, on the importance of clear escalation pathways [39]. Furthermore, VWs contribute to cost savings and reduced emissions by minimizing patient and visitor travel, consistent with recent research advocating for sustainable healthcare models [40].

**Quality of care.** Concerns regarding care quality in VWs focused on safety, consultation limitations, and medication management. Challenges identified included duplicated efforts, time constraints in virtual consultations, and the need for comprehensive diagnostic capabilities. Similar to Chua et al. (2022), our findings highlight the importance of ensuring patient safety, especially when caregivers are remote [41]. A recent public survey indicated patients feel safe receiving care at home [42], yet concerns about home hazards, patient mobility, and lone worker visit persisted. Our review highlights the need for robust safety protocols, including GPS tracking and alarm devices. Greene et al. (2024) noted that virtual assessments may miss the early signs of patient deterioration, though Reid et al. (2021) found that less than 20% of patients required in-person follow-up when virtual assessments were limited [43].

Personalized medication reviews in VWs improved adherence and reduced adverse events, consistent with McGlen et al. (2023) [44] and Kirkcaldy et al. (2018) [45]. Challenges such as limited access to patient health records, e-fax prescription issues, polypharmacy and the risk of antibiotics over-prescription were also identified in this review, underscoring the need for integrated technology solutions.

**Technology and communication.** Successful VW models rely on accessible technology and effective communication. Our findings align with Greene et al. (2024), who emphasized the importance of user-friendly video platforms and integrated EHRs for seamless care transitions. However, they noted that the technology was not always reliable in detecting health deterioration, so reinforcing the need for simplified and improved digital tools [38].

Addressing digital inequality, privacy concerns and cybersecurity is crucial for optimizing VW technology, consistent with other studies [46]. Furthermore, Greene et al. (2024) noted that older individuals who were not skilled with digital technology were not disadvantaged as the care team was flexible in providing individualized support and would offer in-person services. Effective communication within VW teams and between virtual and in-person care providers is essential, supported by studies emphasizing continuous communication and role clarification [41,47–49]. Teams also benefited by finding ways to incorporate informal communication into the workflow and using closed-loop communication to decrease missed communications [48]. Hence, regular meetings, structured handovers, and secure communication platforms can foster collaboration and trust among healthcare teams.

**Training and awareness.** Clear referral guidelines and comprehensive training are essential to optimize VW implementation. Similar to previous studies, we found that interprofessional training and regular updates on service capacity are vital for ensuring appropriate patient referrals and service efficiency [35,47]. Similarly, other studies that reported that a lack of awareness among ED staff was a common barrier to VW adoption, underlining the need for comprehensive training for healthcare providers on virtual care practices [41,50]. Chua et al. (2022) noted that substantial clinical training is needed to equip healthcare professionals to manage acute patients at home [41]. Beyond clinical skills training, teamwork education can foster collaboration and mutual trust among multidisciplinary teams [51]. Furthermore, the role of caregivers is critical for service acceptance, with Norman et al. (2023) emphasizing the need for further evaluation of caregiver outcomes [4]. However, our review did not explore the perspective of caregivers or patients. These findings collectively underscore the importance of targeted training and education to optimize VW implementation and integration into healthcare systems.

## Strengths and limitations

Although we used the Arksey and O'Malley's [13] six-step methodological framework, a few limitations should be considered when interpreting the findings of this review. To minimize the errors, two reviewers identified relevant studies, and we adopted a comprehensive search approach, however, there is a possibility that not all publications relevant to the inclusion criteria were identified by the searches or databases used. We also did not assess the quality of studies, as the focus was on mapping the evidence. Another limitation includes the exclusion of papers that did not directly address acute care in VWs. Many papers simply stated that the intervention included an interprofessional team, and these papers were deemed ineligible for the review. The perspectives of staff are crucial as previous research has suggested that practitioner support can account for a significant portion of the variability in the adoption, growth, and continuity of telehealth services [52]. However, future research could focus on the perspectives of caregivers and patients, particularly in their lived experience of specific VWs [32].

Our search strategy, limited to English-language papers from the last 10 years, may have excluded relevant studies published in other languages or previous years. Most of the articles included in this review were from Western regions (especially UK, Europe, the United States and Australia) with robust healthcare systems. Hence, findings may not be representative of those in other countries and cultures. Additionally, this scoping review examined a diverse set of stakeholders' perspectives (e.g., healthcare professionals, administrators) of virtual wards in different contexts (e.g., rehabilitation, cardiovascular, or frailty units) that may not represent discipline- or disease-specific needs of the stakeholders. Future research could focus on disease specific outcomes from virtual wards. A final limitation is the rapid growth of the field, so it is important to acknowledge that this scoping review is a snapshot at a particular point in time.

## Implications for research, education and practice

Research in this area is evolving rapidly with primary studies becoming available, ensuring that the implementation of services is informed by evidence is a priority [4]. Our scoping review highlights not only the various challenges and considerations but also the significant positive impacts associated with implementing acute virtual wards. Addressing these issues is critical for enhancing the effectiveness of virtual wards as an alternative to traditional hospital care. By understanding and addressing

these challenges, we can further improve patient outcomes and alleviate the pressures on healthcare systems. Future research should continue to explore these challenges to optimize virtual ward models, ensuring they provide an effective alternative to traditional inpatient care. Additionally, continuous evaluation and adaptation of these models will be essential in meeting the evolving needs of patients and healthcare providers, ultimately contributing to a more adaptive healthcare system.

## Conclusion

VWs offer an alternative to traditional inpatient care, with high levels of satisfaction reported by both healthcare staff and patients. However, successful implementation depends on robust service design, clear referral pathways, and the integration of patient and caregiver perspectives. Challenges related to quality of care, technology, and training must be addressed to optimize outcomes. Future research should explore long-term patient outcomes, caregiver experiences, and disease-specific VW models to strengthen the evidence base and inform best practices.

## Supporting information

**S1 Checklist. PRISMA – ScR checklist.**
(DOCX)

## Acknowledgments

We would like to acknowledge support from University College Dublin's (UCD) School of Medicine, College of Health and Agricultural Sciences, and Summer Student Research Awards (SSRA) program. Additionally, we extend our gratitude for support provided to study investigators through the UCD Clinical Research Centre, Ireland East Hospital Group and the Health Research Board.

## Author contributions

**Conceptualization:** Walter Cullen.

**Data curation:** Theresa Sunny, Nandakumar Ravichandran.

**Formal analysis:** Theresa Sunny, Nandakumar Ravichandran.

**Investigation:** Theresa Sunny, Nandakumar Ravichandran, John Broughan, Geoff McCombe, Sheila Loughman, Kenneth McDonald, Neasa Starr, Walter Cullen.

**Methodology:** Nandakumar Ravichandran, Geoff McCombe, Walter Cullen.

**Project administration:** Nandakumar Ravichandran, John Broughan, Geoff McCombe, Walter Cullen.

**Resources:** Walter Cullen.

**Supervision:** Walter Cullen.

**Writing – original draft:** Theresa Sunny, Nandakumar Ravichandran.

**Writing – review & editing:** Theresa Sunny, Nandakumar Ravichandran, John Broughan, Geoff McCombe, Sheila Loughman, Kenneth McDonald, Neasa Starr, Walter Cullen.

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
