## [Decision Letter · Decision Letter 0]

7 Mar 2025

PDIG-D-24-00573Practitioners’ perspectives on implementation of acute virtual wards: A scoping reviewPLOS Digital Health Dear Dr. Ravichandran, Thank you for submitting your manuscript to PLOS Digital Health. After careful consideration, we feel that it has merit but does not fully meet PLOS Digital Health's publication criteria as it currently stands. Therefore, we invite you to submit a revised version of the manuscript that addresses the points raised during the review process. Please submit your revised manuscript within 60 days May 06 2025 11:59PM. If you will need more time than this to complete your revisions, please reply to this message or contact the journal office at digitalhealth@plos.org. Please include the following items when submitting your revised manuscript:* A rebuttal letter that responds to each point raised by the editor and reviewer(s). You should upload this letter as a separate file labeled 'Response to Reviewers '. This file does not need to include responses to any formatting updates and technical items listed in the 'Journal Requirements' section below.* A marked-up copy of your manuscript that highlights changes made to the original version. You should upload this as a separate file labeled 'Revised Manuscript with Track Changes '.* An unmarked version of your revised paper without tracked changes. You should upload this as a separate file labeled 'Manuscript '. If you would like to make changes to your financial disclosure, competing interests statement, or data availability statement, please make these updates within the submission form at the time of resubmission. Guidelines for resubmitting your figure files are available below the reviewer comments at the end of this letter. We look forward to receiving your revised manuscript. Kind regards, Calvin Or, PhDSection EditorPLOS Digital Health Calvin OrSection EditorPLOS Digital Health Leo Anthony CeliEditor-in-ChiefPLOS Digital Healthorcid.org/0000-0001-6712-6626 **Journal Requirements:**

1. We have amended your Competing Interest statement to comply with journal style. We kindly ask that you double check the statement and let us know if anything is incorrect.

2. Figure 2: please (a) provide a direct link to the base layer of the map (i.e., the country or region border shape) and ensure this is also included in the figure legend; and (b) provide a link to the terms of use / license information for the base layer image or shapefile. We cannot publish proprietary or copyrighted maps (e.g. Google Maps, Mapquest) and the terms of use for your map base layer must be compatible with our CC-BY 4.0 license.

* Natural Earth - All maps are public domain. (http://www.naturalearthdata.com/about/terms-of-use/) **Additional Editor Comments (if provided):**- Many of the findings are vague. Please be more specific. For example, “Engagement and active support from hospital specialists (such as cardiology) was also found to be essential in promoting referrals to VW.” What kind of engagement? What type of support? I find the contents in section 4(a), "Referral guidelines," particularly unclear. These are just examples. It would be great if the authors could go through the findings and discussion to identify content and concepts that are too general and present them more specifically.<o:p></o:p>- Please provide a conclusion section to present take-home messages or conclusion statements.<o:p></o:p>**Reviewers' Comments:** Reviewer's Responses to Questions

**Comments to the Author**

1. Does this manuscript meet PLOS Digital Health’s publication criteria ? Is the manuscript technically sound, and do the data support the conclusions? The manuscript must describe methodologically and ethically rigorous research with conclusions that are appropriately drawn based on the data presented.

Reviewer #1: Yes

Reviewer #2: Yes

2. Has the statistical analysis been performed appropriately and rigorously?

Reviewer #1: N/A

Reviewer #2: N/A

3. Have the authors made all data underlying the findings in their manuscript fully available (please refer to the Data Availability Statement at the start of the manuscript PDF file)?

Reviewer #1: Yes

Reviewer #2: Yes

4. Is the manuscript presented in an intelligible fashion and written in standard English?

Reviewer #1: Yes

Reviewer #2: Yes

5. Review Comments to the Author

Reviewer #1: Thank you so much for allowing me to read your work in this area of virtual wards. As you say there is no doubt that the roll-out of virtual wards offers promise in addressing some of the challenge that is facing health systems globally. I wonder if I could ask you to consider;

1. the evidence behind virtual wards is evolving and is less than robust - how is this likely to impact the opinions of health care staff?

2. How representative are healthcare staff who are part of this research review?

3. Given the significant heterogeneity of studies identified and how many are excluded how robust are the conclusions?

4. It is difficult to pick out the key messages in the discussion - please could this be revised?

Reviewer #2: Thank you for the opportunity to review this manuscript. The concept of virtual care and specifically ‘virtual wards’ is evolving and generating greater interest within health care systems. I believe this research is timely and important. The edits and comments are intended to strengthen this work and enhance clarity, accuracy, and reader ease of comprehension.

Line 51- 53 Are you stating that the increase in adults over the age of 80 years is responsible for ER and hospital over-crowding? Please revise with consideration given to the multiple contributors to hospital/ emergency use. This appears to be an ageist perspective on health systems utilization.

Line 55-56 Clarity and greater description would be helpful and would contribute to a more comprehensive understanding of health service challenges

Line 60-63 My understanding is that virtual care also exists within acute care settings and not isolated within home settings. Please confirm and consider revision

Line 70 “fully embed service models” -please provide clarity to this suggestion; are you referring to virtual wards?

Line 74-75 you make a claim that “…VW programme exemplifies the solutions promoted by HSE’s ‘Telehealth Roadmap 2024-27’ strategy - please outline how VWs will address the suggested strategies included in the Roadmap.

Line 78 and throughout the manuscript please use consistent terms or define roles such as “clinician” -how are you defining practitioners / clinicians?

Line 81-82 you state “Studies aimed at understanding the attitudes of those at the centre of implementation have been emphasized as a crucial component of the implementation process…” Consider re-phrasing; most successful implementation projects acknowledge the importance of an interdisciplinary team. Clinicians would be an important stakeholder as the end user.

Line 94-95 Please clarify; do you mean that a study protocol was not generated as a separate manuscript?

Line 142 Additional detail is needed for data analysis; how many researchers analyzed the data; what was your process to facilitate consistent analyses among researchers; how did you resolve differences; describe the background and experience among researchers contributing to the analysis - who had experience working with VW; what disciplines contributed (engineering, nurse, physician, IT...). Was the analysis inductive/deductive or both; did you have an initial code book to direct your analysis; did you use an implementation theory or framework to guide this analysis....

Line 163-164 Please provide clarity and revise; are you suggesting broadly that all staff (who is 'staff'?) reported satisfaction (satisfaction with what). And this was supported by one reference?

Line 164-165 See my comment above; these statements need to be tempered / contextualized

Line 181 “resource allocation” - what resources (do you mean people?)are you referring to?

Line 188-189 How did you know administration was supportive or not? Please review for clarity

Line 204-207 you state: “AVWs offer system-wide benefits, such as reduced emissions, cost savings, and shorter hospital stays (18, 27, 34). However, concerns about climate impact from vehicle use persist, with recommendations for electric vehicles (18). Financial sustainability remains a challenge, requiring public funding and regional support”; these are broad statements that require additional description/ ‘evidence’ for readers to fully comprehend and appreciate the intended.

Line 215 regarding safety and the use of alarm devices resulting in increased confidence; please provide additional insight into ‘lone workers’ and the purpose of home visits within a VW model of care; please address whose confidence is impacted - the health care organization or the provider or both? What is the scope of this concern?

Line 216-217 you state “Technology such as GPS tracking and recording devices may enhance safety”. Please consider the alternative that such tracking may elevate the experience of surveillance among providers or workers or … and this may have implications for enhanced safety but also for lack of privacy.

All results tables – please revise the title to include VW

Line 229-230 would you consider the clinical experience expressed here to be unique to only within VW models of care?

Line 234 you claim ‘…the need for active patients and roles” ; provide additional detail to support reader comprehension of this claim – describe how patients need to be ‘active’ etc.

Line 235 you “…more intimate environment…” ; do you mean their home context or lived environment? Please provide clarity

Line 280-281 requires additional description/ information

Line 311-312 consider caution in the broad statement / claim made here - the literature is limited and this is based on one or few studies

Line 382 “methodological considerations” appears to be your limitations section – please revise.

6. PLOS authors have the option to publish the peer review history of their article (what does this mean? ). If published, this will include your full peer review and any attached files.

**Do you want your identity to be public for this peer review?** For information about this choice, including consent withdrawal, please see our Privacy Policy .

Reviewer #1: No

Reviewer #2: No

---

## [Decision Letter · Decision Letter 1]

14 Apr 2025

Practitioners’ perspectives on implementation of acute virtual wards: A scoping review

PDIG-D-24-00573R1

Dear Mr Ravichandran,

We are pleased to inform you that your manuscript 'Practitioners’ perspectives on implementation of acute virtual wards: A scoping review' has been provisionally accepted for publication in PLOS Digital Health.

Best regards,

Calvin Or, PhD

Section Editor

PLOS Digital Health

**Additional Editor Comments (if provided):**

**Reviewer Comments (if any, and for reference):**

Reviewer's Responses to Questions

**Comments to the Author**

1. If the authors have adequately addressed your comments raised in a previous round of review and you feel that this manuscript is now acceptable for publication, you may indicate that here to bypass the “Comments to the Author” section, enter your conflict of interest statement in the “Confidential to Editor” section, and submit your "Accept" recommendation.

Reviewer #1: All comments have been addressed

Reviewer #2: All comments have been addressed

2. Does this manuscript meet PLOS Digital Health’s publication criteria ? Is the manuscript technically sound, and do the data support the conclusions? The manuscript must describe methodologically and ethically rigorous research with conclusions that are appropriately drawn based on the data presented.

Reviewer #1: Yes

Reviewer #2: Yes

3. Has the statistical analysis been performed appropriately and rigorously?

Reviewer #1: N/A

Reviewer #2: N/A

4. Have the authors made all data underlying the findings in their manuscript fully available (please refer to the Data Availability Statement at the start of the manuscript PDF file)?

Reviewer #1: Yes

Reviewer #2: Yes

5. Is the manuscript presented in an intelligible fashion and written in standard English?

Reviewer #1: Yes

Reviewer #2: Yes

6. Review Comments to the Author

Reviewer #1: Thank you so much for revising this important work and I agree that this is a key area for successful implementation

Reviewer #2: Suggested edits have been addressed.

7. PLOS authors have the option to publish the peer review history of their article (what does this mean? ). If published, this will include your full peer review and any attached files.

**Do you want your identity to be public for this peer review?** For information about this choice, including consent withdrawal, please see our Privacy Policy .

Reviewer #1: No

Reviewer #2: No
